# Effects on the Properties after Addition of Lithium Salt in Poly(ethylene oxide)/Poly(methyl acrylate) Blends

**DOI:** 10.3390/polym12122963

**Published:** 2020-12-11

**Authors:** Suhaila Idayu Abdul Halim, Chin Han Chan, Jörg Kressler

**Affiliations:** 1Faculty of Applied Sciences, Universiti Teknologi MARA, Shah Alam 40450, Selangor, Malaysia; suhailaidayu@gmail.com; 2Faculty of Natural Sciences II, Institute of Chemistry, Martin-Luther University Halle-Wittenberg, 06120 Halle, Germany; joerg.kressler@chemie.uni-halle.de

**Keywords:** poly(ethylene oxide), poly(methyl acrylate), phase morphology, dielectric relaxation

## Abstract

The studies of phase behavior, dielectric relaxation, and other properties of poly(ethylene oxide) (PEO)/poly(methyl acrylate) (PMA) blends with the addition of lithium perchlorate (LiClO_4_) were done for different blend compositions. Samples were prepared by a solution casting technique. The binary PEO/PMA blends exhibit a single and compositional-dependent glass transition temperature (*T*_g_), which is also true for ternary mixtures of PEO/PMA/LiClO_4_ when PEO was in excess with low content of salt. These may indicate miscibility of the constituents for the molten systems and amorphous domains of the systems at room temperature from the macroscopic point of view. Subsequently, the morphology of PEO/PMA blends with or without salt are correlated to the phase behavior of the systems. Phase morphology and molecular interaction of polymer chains by salt ions of the systems may rule the dielectric or electric relaxation at room temperature, which was estimated using electrochemical impedance spectroscopy (EIS). The frequency-dependent impedance spectra are of interest for the elucidation of polarization and relaxation of the charged entities for the systems. Relaxation can be noted only when a sufficient amount of salt is added into the systems.

## 1. Introduction

Lithium (Li)-ion batteries that exhibit appreciably high energy and power density are the choice of the electrochemical energy storage for portable electronics/devices, hybrid/full electric vehicles, etc. [1,2,3]. Li-ion batteries are rechargeable devices, where both electrodes are intercalation materials, and the commercial electrolyte is commonly Li salt dissolved in a mixture of organic solvents [3,4,5]. Extensive studies on organic solvent-free electrolytes, for example on solid polymer electrolytes (SPEs), have been carried out [6,7] since 1980. One of the popular polymer hosts for SPEs is poly(ethylene oxide) (PEO), which is a semi-crystalline polymer with crystallinity amount up to 70% [8,9,10,11,12]. It is widely accepted that ionic percolation mainly takes place in the amorphous regions of polymer [6,11].

One of the strategies to enhance the ionic conductivity (*σ*_DC_) of binary mixtures of PEO and Li salt is to add a second polymer that is miscible or homogenous with PEO (in the melt or amorphous phase), for example poly(methyl acrylate) [9,13], poly(methyl methacrylate) [14], polyacrylate (PAc) [15], poly(propylene oxide) (PPO) [16], poly(ε-caprolactone) (PCL) [17,18], etc., with the aim to suppress the crystallinity of PEO. However, suppression of crystallinity in PEO alone does not always lead to enhancement of conductivity for the systems as compared to the PEO–salt systems [19]. This is due to highly complex systems that develop when Li salt is added to the miscible or homogenous PEO (binary) blends at room temperature, let say at 25 °C, where the conductivity is measured for useful applications. Therefore, the salt content, blend compositions, the glass transition temperature (*T*_g_) of the second polymer, and homogenous or heterogeneous amorphous phase of the ternary systems will affect the conductivity of the materials.

In this study, high molar mass PEO is blended with an amorphous polymer, i.e., PMA, with the *T*_g_ roughly at 10 °C, and with addition of lithium perchlorate (LiClO_4_). PEO/PMA blends are known to be homogenous above melting point of PEO (~65 °C) and in the amorphous phase for all blend compositions under the experimental condition [13,20]. Upon cooling from the melt, the PEO starts to crystallize and phase separates from the mixture. Hence, there is co-existence of PEO crystalline phase and the amorphous mixture (amorphous PEO and PMA) at 25 °C (room temperature).

The addition of the ternary component, the Li salt to PEO/PMA blends may trigger phase separation of PEO and PMA in the amorphous phase of the system depending on the composition of the blends and the salt content [13]. Hence, the amorphous phase of the ternary systems can be homogenous or heterogeneous (c.f. text below) depending on the composition of the blends as well as the salt content, which in the end may lead to enhancement or reduction on conductivity as compared to the binary PEO–salt system at a constant mass fraction of salt (*W*_S_) or at constant mass fraction of PEO (*W*_PEO_). It is noteworthy that these polymer electrolyte systems are becoming more complex and complicated in the ternary mixture at 25 °C.

As a result, these composition-dependent ternary systems of PEO/PMA/LiClO_4_ will have different morphologies, which lead to different conductivities at constant salt content or constant blend composition. The composition-dependent morphologies of the ternary mixtures in the amorphous region are listed in Table 1 as an overview.

Situation (I) is based on the existence of two distinct phases (liquid–liquid separation) in the system and situation (II) assume two amorphous phases of PEO and PMA incorporated with Li–salt exist as a single phase. However, another possibility may emerge in the binary and ternary systems with much higher concentration of salt, where the precipitation of Li–salt may be observed out of the polymer matrix when the mixtures are up to saturation, i.e., situations (III) and (IV) [21]. Situation (IV) considers the two amorphous phases of PEO and PMA are no longer homogeneous (miscible) as in (II), along with the presence of pure salt (salt precipitation) after the addition of sufficiently high salt content or at blend composition with minor PEO. Morphology with situation (I) might be effective for conductivity enhancement, as it may increase the distribution of Li–salt into the preferred polymer phase, consequently increasing the polymer–salt interaction. In short, all the morphologies discussed above may play a vital part to the electrical phenomenology of PEO/PMA SPEs. Thus, we note here that frequency-dependent impedance studies are deemed important to elucidate the dielectric and electric relaxation of dipolar entities in the polymer electrolytes as impedance (*Z*) is one of the central quantities in impedance spectroscopy.

This work is an extension of the previous contribution to the solid polymer electrolytes (SPEs). The scope of the discussion of these ternary PEO/PMA/LiClO_4_ systems were expanded with in-depth theoretical analyses and additional experiments as compared to the previous study [13]. Differential scanning calorimetry (DSC) was employed to estimate the properties of the amorphous phase (on the *T*_g_) as well as the crystalline phase (on the melting behavior and its crystallinity) of the ternary systems. Morphology and dielectric response of the systems were investigated by optical microscopy (OM) and electrochemical impedance spectroscopy (EIS), respectively. Besides, polarization and relaxation of dipoles in composition-dependent PEO/PMA blends after addition of Li–salt was studied by EIS and will be discussed.

## 2. Experimental

### 2.1. Materials

Characteristics of the polymers and salt are given in Table 2. The polymers were purified prior to further preparation. PEO was purified by dissolution in chloroform (Merck, Darmstadt, Germany) following with precipitation in *n*-hexane (Merck, Darmstadt, Germany). The as-received PMA in toluene was precipitated in *n*-hexane before blending. The LiClO_4_ was dried at 120 °C for at least 24 h.

### 2.2. Preparation of Samples

Binary blends of PEO/PMA were prepared by using a solution casting technique. Solid film PEO_80_ denotes PEO/PMA 80/20 blend (*m*/*m*) and analogue sample coding for other compositions. Quantity *m_i_* represents mass of component *i*. The solid solution comprising of PEO, PMA, and LiClO_4_ were also prepared by solution casting technique. The mass fraction of polymer (PEO) and mass fraction of salt were estimated as below:(1)$$W_{\mathrm{PEO}}=\frac{m_{\mathrm{PEO}}}{m_{\mathrm{PEO}}+m_{\mathrm{PMA}}+m_{S}}\;\mathrm{and}\;W_{S}\frac{m_{S}}{m_{\mathrm{PEO}}+m_{\mathrm{PMA}}+m_{S}}$$

Corresponding *W*_S_ to mole fraction of salt (*X*_S_) and salt content (*Y*_S_) is shown in Table A1 in Appendix A. All the components were dissolved in acetonitrile (ACN) and stirred for 24 h at 50 °C. The solution was casted onto Teflon^®^ dish and left to dry at room temperature until the solvent evaporated. The samples were oven dried at 50 °C for no less than 24 h for removal of residual solvent. This was followed by heating the samples at 80 °C for ½ h under nitrogen atmosphere. Subsequently, the samples were isothermally crystallized at 25 °C for 24 h in the convection oven before vacuum drying at 25 °C for 24 h. All samples were then kept in desiccators at 25 °C. Then, the samples were again vacuum dried at 25 °C for 24 h prior to any characterization.

The solution-cast samples were thermally treated under inert atmosphere above the melting temperature (*T*_m_) of PEO for a certain period of time to erase the thermal history of the sample during the sample preparation pathway as well as for complete mixing of all components in the melt. The consistency in sample preparation is crucial for the reproducible properties reported here. Hence, the electrolyte systems discussed herein are close towards equilibrium condition.

## 3. Characterization

### 3.1. Differential Scanning Calorimetry (DSC)

The quantities *T*_g_, *T*_m_, and melting enthalpy (Δ*H*_m_) of samples were analyzed from the heating cycle of DSC. The samples were studied using DSC TA Q200 (TA Instrument, New Castle, DE, USA) equipped with RCS90 cooling system (TA Instrument, New Castle, DE, USA). Nitrogen gas was purged during analysis at a rate of 50 mL min^−1^ to avoid thermo-oxidative degradation of samples. Roughly 10 to 15 mg of thin film samples were encapsulated in aluminum DSC sample pans for analysis. Calibration of DSC using high-purity indium standard was done prior to analysis. Samples were heated up from −90 °C to 80 °C at a heating rate of 10 °C min^−1^. The *T*_g_ was estimated at half extrapolated change in heat capacity (Δ*C*_p_) or adopted from Moynihan’s approach (if relaxation endotherm overlaid the glass transition) for estimation of *T*_g_ in a more precise manner [21]. *T*_m_ was extracted as the maximum of the endothermal peak in the DSC trace. Δ*H*_m_ was estimated from the area underneath the melting endotherm. The crystallinity (*X*^*^) of the PEO phase in the blends was estimated by Equation (2).
(2)$$X^{*}=\left(\frac{∆H_{m}}{∆H_{\mathrm{ref}}\;\cdot \;W_{\mathrm{PEO}}}\right)\;\times 100\%$$
where Δ*H*_ref_ = 188.3 J g^−1^ is the melting enthalpy of 100% crystalline PEO, and *W*_PEO_ is the mass fraction of PEO in the blends.

### 3.2. Optical Microscopy (OM)

The morphology of each sample was captured using AxioVision Control software (Zeiss, Oberkochen, Germany) connected to the Axioplan 2 imaging polarizing optical microscope (Zeiss, Oberkochen, Germany) equipped with a Linkam TM600/s hotstage (Linkam, Surrey, UK). Three percent (*m*/*v*) of the sample was dissolved in acetonitrile and was heated at 50 °C for 24 h. After dissolution, the polymer solution was casted drop-by-drop on top of a glass cover slip and allowed to dry at room temperature for at least 24 h. The sample was heated up at 80 °C and was annealed for ½ h, followed by quenched cooling and isothermal treatment at *T*_c_ = 25 °C until complete crystallization. Minimum 5 micrographs were captured with 10× magnification at two different temperatures *T* = 80 °C (with non-polarized mode) with annealing time of ½ h to give sufficient time for complete mixing in the melt and at *T* = 25 °C (with polarized mode) with annealing time of 24 h for complete crystallization of PEO.

### 3.3. Impedance Spectroscopy (IS)

Impedance measurement of each sample was done at 25 °C using a Hioki 3532-50 Hi Tester impedance analyzer (Hioki, Chubu, Japan) equipped with a computer for data collection over the frequency range from 50 Hz to 2 MHz. Two stainless steel electrodes with a diameter of 20 mm were used as the current collector and the blocking electrode for the ions. The sample was placed in between the two blocking electrodes for measurement. The *σ*_DC_ value was estimated from the bulk resistance (*R*_b_) following equation of *σ*_DC_ = *L*/(*A*·*R*_b_), where quantities *L* and *A* denote thickness of the sample and surface area in touch with the two stainless steel disc electrodes, respectively. The quantity *L* was measured with Digimatic Caliper (Mitutoyo, Kanagawa, Japan) at three different spots that were in contact to the electrodes. The quantity *L* was averaged from the three measurements and the thickness of the dried samples is maintained within the range of 0.25–0.35 mm.

*σ*_DC_ was either estimated from both Nyquist plot and from the real (*Z*’) or imaginary parts (*Z*″*)* values of impedance at frequency for fully stabilized network ($f_{\max}^{Z^{\prime \prime }}$) [*Z*’($f_{\max}^{Z^{\prime \prime }}$) and *Z*″($f_{\max}^{Z^{\prime \prime }}$)] at maximum of *Z*″, and each result obtained by both methods is maintained with an error of less than or equal to 5%. Values of *σ*_DC_ reported here were the averages of three impedance analyses from three different spots of the thin sample with errors of *σ*_DC_ approximately at 10%.

## 4. Results and Discussion

### 4.1. Glass Transition Temperature

Glass transition temperature (*T*_g_) may be seen as one criterion of the miscibility of a polymer blend from a macroscopic point of view. The blend is seen as a single-phase polymer blend (miscible blend) when it exhibits a single and composition-dependent *T*_g_. Immiscibility can be concluded when two *T*_g_s that are approximately close to the *T*_g_ of parent polymers are observed. The Fox equation as in Equation (3) is commonly referred to for the evaluation or prediction of the *T*_g_ values of miscible binary blends.
(3)$$\frac{1}{T_{g}}=\frac{W_{A}}{T_{g,\;A}}+\frac{W_{B}}{T_{g,\;B}}$$
where *W*_A_, *T*_g_,_A_ and *W*_B_, *T*_g_,_B_ represent the mass fraction and *T*_g_ of the respective polymer. The dashed curve in Figure 1b is calculated after Equation (3). The entire composition of semi-crystalline/amorphous PEO/PMA blends in this study exhibit only a single and composition-dependent *T*_g_. The heating cycle of DSC thermograms are displayed in Figure 1a. Moreover, the experimental *T*_g_ values are in good agreement with the *T*_g_s predicted after the Fox equation. This may suggest that the polymer pair is miscible and homogeneous in the amorphous phase as well as in the molten state under this experimental condition for all compositions. 

Figure 2a displays that the *T*_g_ of PMA decreases with elevating salt content (*W*_S_). Unlike PMA, *T*_g_ of neat PEO elevates with increasing *W*_S_. This phenomenon implies that the salt may be more soluble in PEO than PMA. It is interesting to note that the miscible PEO/PMA blends act differently with the addition of salt as shown in Figure 2. The blends with *W*_PEO_ ≥ 0.7 after the addition of salt *W*_S_ still show single-composition-dependent *T*_g_ for all studied salt concentrations (*W*_S_ = 0–0.17), which may indicate miscibility of the ternary mixtures in the amorphous phase and in the molten state. PEO_80_ displays the highest *T*_g_ values at *W*_S_ = 0–0.09 as compared to other systems *W*_PEO_ ≥ 0.7 with the same amount of salt. It implies PEO_80_ may be an effective host for LiClO_4_ for the enhancement of the ionic conductivity. Normally, the increase in *T*_g_ indicates the stiffness of the polymer chains at an increasing temperature. In this case, the stiffness of the chain segments comes from the interaction of salt molecules and polymer chains. However, the increase in *T*_g_ is only limited in the range of low salt content, as it increases up to saturation of the mixture due to the solubility limit of salt in the respective polymers [23,24]. When *W*_PEO_ ≤ 0.6 at *W*_S_ ≥ 0.05, the heterogeneity in the melt is inferred (i.e., liquid–liquid phase separation). It is deduced by the presence of two *T*_g_s that correspond to the parent polymers. From Figure 2b, another observation on the salt localization in the PEO phase more than PMA can be seen. In the immiscible systems, the *T*_g_s of PEO are constantly observed at slightly higher values than the neat PEO, whereas the *T*_g_s of PMA lie closely to that of the neat PMA.

These *T*_g_ findings are in good agreement with the descriptions in the Introduction. Macroscopically, we observe situation (I) in the system with *W*_PEO_ ≤ 0.6 at *W*_S_ ≥ 0.07, and situation (II) in the system of *W*_PEO_ ≥ 0.7 at *W*_S_ ≤ 0.10. One may also observe the morphology of situation (III) in binary PEO–salt systems (*W*_PEO_ = 1) at *W*_S_ ≥ 0.10 or beyond the salt saturation of the mixture. This is confirmed by the *T*_g_ of the respective mixture, which is closed to *T*_g_ of neat PEO. Furthermore, the precipitation of Li–salt was observed from optical inspection. The mixture with morphology (III) can be differentiated into salt-rich or salt-poor phases [21]. The salt-rich phase (phase″) is mainly the pure salt phase, and salt-poor phase (phase’) is the phase of highly diluted salt solution in the polymer phase. The salt content of salt-poor phase (*W*_S_’) can be elucidated instantly from the plot of *T*_g_ against *W*_S_. Hence, the preferential localization of salt in the different polymer phases of the immiscible ternary mixture can be detected based on their respective *T*_g_ values. For instance, the PEO_50_ system added with *W*_S_ at 0.091 (black arrow in Figure 2b) is seen to contain approximately 1 wt.% of Li–salt dispersed in the PEO phase as the *T*_g_ of PEO in PEO_50_ is equivalent to *T*_g_ of PEO at *W*_S_ = 0.01 (black arrow in Figure 2a).

These ternary mixtures are miscible in the molten state and in the amorphous state under this experimental condition at sufficiently low salt content for all blend compositions. The mixtures are also miscible in molten state and in the amorphous phase at high salt content only when the PEO content is in excess, *W*_PEO_ ≥ 0.7. The liquid–liquid phase separation in the molten state may be deduced when the PEO content is *W*_PEO_ ≤ 0.6 at high salt content, *W*_S_ ≥ 0.05. The mixtures under discussion are comprised semi-crystalline/amorphous PEO/PMA SPEs. Hence, at 25 °C, the SPEs (which is true either for miscible or liquid–liquid phase-separated systems in the melt) are seen as liquid–solid phase separated systems due to the crystallization of PEO, which takes place upon cooling from the melt. The morphological studies of these SPEs that will be discussed in the subsequent section were carried out at temperatures *T* = 80 °C and *T* = 25 °C using optical microscopy (OM) with the aim of elucidating the phase behavior of the mixture in the melt and in the liquid–solid form from the microscopic level aside from distinguishing the correlation between *T*_g_ and morphology. However, the correlation between the electrical properties and miscibility will not be attempted because only liquid–solid phase-separated systems at 25 °C were studied in all cases. Instead, preferential localization of salt in different polymer phases and the percolation path of the systems will be correlated to the relaxation and conductance of the systems. 

From Figure 2a, we may also discuss the variation of *T*_g_ of the polymer blends that are still miscible in the molten and amorphous state after the addition of salt, for instance, *W*_PEO_ ≥ 0.7, in terms of thermodynamics based on the empirical observation [21]. The quantity of *T*_g_, and its Δ*C*_P_ can be correlated as follows
(4)$$\ln\;\left(\frac{T_{g}}{T_{g}^{o}}\right)=\;\frac{-∆S^{\;\mathrm{glass}}}{∆C_{P}}$$

Δ*S*^glass^ denotes the change in entropy of the glassy state after addition of salts that indirectly describes the deviation of the glass from the equilibrium. Equation (4a) illustrates the Δ*C*_P_ serves as the factor of proportionality of –Δ*S*^glass^ and *T*_g_/*T*_g_^o^. It is also true in the small range of *T*_g_^o^ to *T*_g_ that Δ*C*_P_ is assumed to be independent of temperature (this observation is only applicable for thermodynamically miscible system, i.e., PEO–PEO_70_ and PMA). Approximately, the quantity of *T*_g_/*T*_g_^o^ may also be expressed as below
(5)$$\frac{T_{g}}{T_{g}^{o}}=1+\frac{1}{T_{g}^{o}}\left(\frac{\partial T_{g}}{\partial W_{S}}\right)W_{S}$$
where the slope of plot *T*_g_ against *W*_S_ reads, $\frac{1}{T_{g}^{o}}\left(\frac{\partial T_{g}}{\partial W_{S}}\right)$ ≡ *Γ*. The relationship of Equations (4) and (5) explains that the slope *Γ* is interconnected to the partial molar entropy of the polymer pair in the ternary mixture that can be defined as $-∆\tilde{S}\equiv \frac{\partial ∆S^{\mathrm{glass}}}{\partial W_{S}}\equiv \Gamma \;\bar{∆C_{P}}$. The mean value of Δ*C*_P_ of the polymer pair with different salt content is employed for estimation of $\Delta \tilde{S}$ on condition that the ternary mixtures lies in the linearity. Furthermore, the glass transition is not accompanied by the change in energy; hence, we may formulate the related chemical potential or molar partial Gibbs free energy by
(6)$$∆G=\;-T_{g}^{o}∆\tilde{S}$$

The relationship of (4) and (5) shows when the slope *Γ* is positive; the reduced *T*_g_/*T*_g_^o^ of the mixture should be larger than unity and Δ*S*^glass^ would be negative. The positive gradient, *Γ* > 0 reflects the freezing in of degrees of freedom of the polymer chains, whereas a negative gradient, *Γ* < 0 shows the increase in the chain’s degrees of freedom of the polymer. Experimentally, we observe that the salt affects the respective binary polymer-salt mixtures differently. PEO has a positive slope, *Γ* > 0 with increasing *W*_S_, whereas PMA has a negative slope, *Γ* < 0 with increasing *W*_S_. It suggests that the addition of salt in PEO leads to a higher stiffness of polymer chains whereas salt in PMA increases the flexibility of polymer chains. This discussion is also relevant for miscible ternary mixtures of PEO/PMA/LiClO_4_. The gradient *Γ* of *W*_PEO_ ≥ 0.7 (i.e., PEO_90_ and PEO_70_) is reduced gradually with the increasing content of PMA. This indicates that addition of PMA in the PEO–salt systems increases the degree of freedom of the polymer chains in the miscible polymer blends (increasing the chain flexibility). Besides, it is worth noting that addition of certain amount of PMA into the PEO–salt system also drives the systems closer to equilibrium, as reflected in the Δ*G*. The corresponding parameters *Γ* of miscible PEO/PMA/LiClO_4_ mixtures are summarized in Table 2.

### 4.2. Crystallinity

The degree of crystallinity (*X**) of PEO in the ternary mixtures was estimated from Equation (2). The melting enthalpy (Δ*H*_m_) of PEO was obtained from the area under the melting endotherm of the thermograms. Δ*H*_m_ is an essential quantity for the estimation of the degree of crystallinity (*X*^*^) for semi-crystalline polymer as the extent of the liquid–solid phase separation can be evaluated from crystallinity. The degree of crystallinity allows for rough estimation of the amorphous content. Hence, in principle, the reduction in crystallinity of the crystalline phase leads to an increase in the amorphous phase. It is widely accepted that ion percolation (conductivity) takes place only in the amorphous phase, thus, a rough estimation of the composition of the amorphous phase might be useful to understand electric conductivity. Equation (2) describes the degree of *X*^*^ of PEO, which is directly proportional to Δ*H*_m_ of PEO, if the enthalpy of 100% crystallinity of PEO serves as the factor of proportionality
*X*^*^ ∝ Δ*H*_m_(7)

Figure 3 depicts the *X*^*^ of PEO in the ternary mixtures as a function of *W*_PEO_. The dashed curve in the plot of *X*^*^ against *W*_PEO_ is the constancy curve of crystallinity of PEO in the ternary mixtures. The suppression of *X*^*^ from the constancy curve may be caused by the entrapment of the other component in the crystalline phase (in this case, PEO crystalline). Figure 3 illustrates that the *X*^*^ of salt-free high molar mass PEO is 70% and starts to gradually level off to ~20% with increasing of salt content up to *W*_S_ = 0.167 for binary PEO–salt system. The deviation from the PEO crystallinity constancy curve is observed for all systems with different blend compositions and salt content.

When PEO crystallinity in both salt-free and salt-added systems is at low salt content, *W*_S_ ≤ 0.05 displays no significant difference. The further suppression of *X*^*^ of PEO is observed with higher content of salt, *W*_S_ ≥ 0.07, which is true for all blends under discussion. This indicates that with small amounts of salt in the mixture, the PEO crystallinity of the semi-crystalline/amorphous mixture is not really affected by the presence of salt and the further suppression of *X*^*^ only starts with increasing amounts of salt in the mixture. These observations are true for both miscible and immiscible systems of this study. The suppression of crystallinity is normally true for miscible systems, as the crystallization behavior of the crystalline component in the immiscible systems (or liquid–liquid phase separated systems in the melt) is expected to have close approximation to the neat crystalline polymer, and it is normally unaffected by the presence of other components. Nevertheless, from this study, we note that salt and PMA are fond of PEO. Hence, we may see the exclusion of salt and PMA from PEO spherulites is not completely perfect, and the entrapment of amorphous phase of PEO, PMA, and salt in PEO spherulites lead to suppression of PEO crystallinity as well as depression of apparent melting temperature of PEO in the systems, which will be discussed in the next section.

### 4.3. Melting Behavior

The apparent melting temperature (*T*_m_) was obtained from the maximum of the melting endotherm of the DSC curve. Figure 4 depicts the apparent *T*_m_ of PEO in the ternary mixtures as a function of the mole fraction of salt (*X*_S_). As mentioned before, the decrease in the crystallinity of the crystalline phase is generally influenced by the miscibility of the mixture, which is also true for the apparent *T*_m_. The depression of the apparent *T*_m_ of PEO is expected in miscible systems. We note here, both melting point and enthalpy are greatly depending on the amount of salt content. The apparent *T*_m_s of PEO of the as-prepared samples are depressed significantly from the constancy curve with increasing salt content for all compositions. This implies that the dissolution of Li–salt in the polymer matrix (in PEO) increases, which is in good approximation to the *T*_g_ as well as the crystallinity findings.

For binary polymer–salt mixtures with a crystalline constituent, we may employ the melting point depression of the crystalline polymer for estimation of deviations from perfect behavior [25,26]. In this case, when binary PEO/Li mixtures are completely miscible in the molten state, we employ the melting point depression of PEO for evaluation of the solid solution from perfect behavior (i.e., PEO–PEO_70_ at *W*_S_ ≤ 0.10). The estimation is expressed by
(8)$$T_{m}=T_{m}^{o}+\frac{R{\left(T_{m}^{o}\right)}^{2}}{∆H_{\mathrm{ref}}}\left(\frac{\ln\;\gamma_{P}}{{\ln\;X}_{P}}+1\right){\ln\;X}_{P}$$
where *X*_P_ denotes the mole fraction of PEO in the molten state, and *γ*_P_ is the corresponding activity coefficient for the solution, whereas the quantities *T*_m_^o^ and ∆*H*_ref_ symbolize the melting temperature of neat PEO and the melting enthalpy of 100% crystallinity of PEO, respectively. For systems with low salt content (or the completely miscible systems, PEO–PEO_70_ at *W*_S_ ≤ 0.10), we may assume ln *X*_P_ = −*X*_S_. Hence, we see from Equation (8), the information about the activity coefficient *γ*_P_ may be determined instantly from the linear plot of *T*_m_ against *X*_S_. This relationship is valid as long as the PEO crystalline phase crystallizes out from the molten mixture upon cooling from the melt. We apply Equation (8) to the as-prepared PEO/Li mixture for estimation of deviations from the ideal behavior (c.f. red-dashed curve in Figure 4). From Figure 4, we obtained ∆*T*/K = 147.3 *X*_S_ for the PEO/Li mixture. The deviation of melting point depression from perfect behavior [∆*T*(*X*_S_)] for the PEO/Li mixture, specifically at higher salt content (*X*_S_ > 0.05), is observed. This indicates at low salt contents that the melting point depression is not influenced by the salt, and it shows to a good approximation perfect behavior, displaying a very small *γ*_P_ = 1 + 0.30 *X*_S_. This observation is also in good agreement with the findings discussed in references [26,27], where the estimation of the melting point depression of the high molar mass PEO (*M*_η_ = 3 × 10^5^ g mol^−1^) against *X*_S_ under similar experimental conditions yields *γ*_P_ close to unity (1 + 0.38 *X*_S_) as reported in reference [27]. This indicates that the PEO/Li system under these experimental conditions behaves nearly perfectly. Similar trend observations are found for other miscible mixtures, such as PEO_90_, PEO_80_, and PEO_70_, where drastic deviations only occur for systems with higher content of salt (*X*_S_ > 0.05). This implies that all systems that are still miscible with low salt contents under these experimental conditions behave nearly perfect as in binary PEO/Li systems. The results retrieved from Equation (8) are summarized in Table 3.

### 4.4. Optical Microscopy

From *T*_g_ measurements, we note the existence of single *T*_g_s for binary PEO/PMA blends, which is also true for ternary PEO/PMA/LiClO_4_ mixtures at sufficiently small salt concentration under described experimental conditions. The heterogeneity (immiscibility) is observed only when the mass fraction of PEO is minor, *W*_PEO_ ≤ 0.6, and the mass fraction of salt is high, *W*_S_ ≥ 0.05. The optical micrographs of selected ternary PEO/PMA/LiClO_4_ systems taken at 10× magnification is shown in Table 4. The micrographs were captured at two different temperatures, i.e., 80 °C after 30 min (in the molten state) and 25 °C after 24 h (in the solid state). As expected, the binary PEO/PMA blends in the molten state (*T* = 80 °C) display only a single-phase without any visible boundary with polymer interfaces. This observation is also true for ternary mixtures with small concentration of salt (i.e., systems of PEO, PEO_80_, and PEO_50_ without the presence of salt). The two-phase structure occurs when the mass fraction of PEO is minor, *W*_PEO_ ≤ 0.6, and mass fraction salt is high, *W*_S_ ≥ 0.05. It is supported by the micrographs of PEO_50_ after addition of *W*_S_ ≥ 0.05, where a two-phase structure with clear boundaries at the polymer interface is observed. PEO_50_ with *W*_S_ = 0.065 displays a matrix-droplet morphology, and it starts to transform into co-continuous morphology with addition of higher salt content at *W*_S_ = 0.091. This observation on PEO_50_ with salt may be associated to the situation (I) and (IV) as described in the introduction, where the liquid–liquid phase separation is inferred in the melt and amorphous phase of the semi-crystalline/amorphous PEO/PMA blends after addition of higher content of salt (this is in good agreement to the *T*_g_ results; system with *W*_PEO_ ≤ 0.6 at *W*_S_ ≥ 0.05, which possess two *T*_g_s that correspond to PEO and PMA).

At 25 °C, where PEO is crystalline below the melting point *T*_m_ = ~65 °C, the samples are placed between crossed polarizers of the optical microscope. Using polarized light, the neat PEO displays large spherulites with a fine fibrillar texture and a clear Maltese cross. As expected, the number of nucleation sites of PEO spherulites for PEO, PEO_80_, and PEO_50_ increase with and without salt due to the exclusion of PMA and Li–salt from the PEO crystalline phase during crystallization This is also reflected by the *T*_m_ results, where the melting point depression of PEO is observed with increasing salt content. This is due the fact that PMA and Li–salt most likely have attractive interactions with PEO phases, which eventually lead to the incomplete exclusion. Coarsening and irregularity of PEO spherulite fibrils with blurred grain boundary are clearly noted for all systems at high salt content. There are several spherical dark spots of PMA within the continuous phase of PEO spherulites for PEO_50_ at *W*_S_ = 0.065 and 0.091.

The proposed schematic diagrams of PEO_50_ systems at *W*_S_ = 0 and 0.091 in the molten and solid state based on the *T*_g_ results and polarized optical microscopy are shown in Figure 5. We conclude that both systems possess different morphologies molten at 80 °C or isothermally crystallized at 25 °C. Immiscibility caused by liquid–liquid phase separation (at 80 °C) (c.f. Figure 5c) and liquid–solid phase separation in PEO_50_ are observed while cooling down from 80 °C to 25 °C (c.f. Figure 5d) after addition of salt *W*_S_ = 0.091. Figure 5 was assigned with the possible morphologies from Table 1.

### 4.5. Impedance Spectroscopy

The impedance spectra of all systems were measured at 25 °C. Selected binary systems of PEO, PMA, and its blends, such as PEO_90_, PEO_80_, PEO_70_, PEO_60_ and PEO_50_, added with different salt content will be discussed under this section, where the morphologies of these systems are correlated to situation (I), (II), (III), or (IV). It is widely accepted that the percolation network only bounds in the amorphous domain of salt-added systems. This phase is considered as not in thermodynamic equilibrium [28,29,30,31]. It may be adequate to discuss the dielectric relaxation process in this domain as a fluctuation–dissipation process [32]. Hence, in the following, the development of relaxation or polarization of dipoles are discussed from the phenomenological point of view.

We observe here that the real part (*Z*’) and imaginary part (*Z*″) of impedance reflect to Ohmic resistance and non-Ohmic resistance, respectively. The quantity *Z*″ displays the characteristic frequencies mainly for dielectric (or dipole) relaxation resulting from local motions of the charged entities and electrode polarization developing from the accumulation of charged entities at the electrode–electrolyte interfaces, after being subjected to an external electric field. These characteristic frequencies are noted as $f_{\max}^{Z^{\prime \prime }}$ and $f_{\min}^{Z^{\prime \prime }}$, respectively. Besides, the intersection between *Z*’ and *Z*″ that is noted as $f_{\mathrm{cross}}^{Z^{\prime }-Z^{\prime \prime }}$ indicates to the development of the percolation charge entities. We recognize the characteristic frequencies as the average of relaxation time constant (*ω*^−1^). A system with one relaxation time (*ω*^−1^) is recognized as Debye relaxation [33]. In general, these phenomena can be observed in a system with highly resistive behavior. A system with high capacitive behavior normally behaves contrarily. Apparently, the impedance plot *Z*″ = *Z*″ (*Z*’) in the *Z*’-*Z*″ plane results in semicircle with a radius of *R*_b_/2. Hence, the plot of *Z*″ vs. frequency (*f*) as displayed in Figure 6 allows for the determination of the bulk resistance (*R*_b_), with |*Z***^″^**_max_| = *R*_b_/2. This *R*_b_ quantity is defined as inversely proportional to the DC conductivity (*R*_b_ ∝ 1/*σ*_DC_). Hence, investigation of the relaxation process in terms of impedance from phenomenological perspective may implicitly elucidate the dielectric and electrical behavior of the system.

Figure 6 represents the impedance spectra, *Z*’ (*f*) and *Z*″ (*f*), of PEO at low salt content *W*_S_ = 0.0196. We observe the existence of the three characteristic frequencies as mentioned before in the impedance spectra of the system. The $f_{\max}^{Z^{\prime \prime }}$ is noted as the main dipole relaxation, which originates from the alignment of dipoles resulting from the interplay of the sample’s resistance and capacitance in the fully stabilized network. The $f_{\min}^{Z^{\prime \prime }}$ is noted as the electrode polarization (or called a double-layer capacitance due to the accumulation of charges at the interface of electrode and electrolyte) especially at low frequency. This observation frequently leads to dispersion of relaxation times (i.e., $f_{\max}^{Z^{\prime \prime }}$ < $f_{\mathrm{cross}}^{Z^{\prime }-Z^{\prime \prime }}$) and normally becomes severe in the system with high salt content because of the inhomogeneity in the system (c.f. red triangle markers for PEO–salt in Figure 7). However, the distance of $f_{\max}^{Z^{\prime \prime }}$ and $f_{\mathrm{cross}}^{Z^{\prime }-Z^{\prime \prime }}$ for PEO at *W*_S_ = 0.0196 is small which may indicate that the system is close to Debye relaxation. We suggest that the orientation of dipolar entities in PEO–salt is dominantly restricted to short-range motion (local motion) and insignificant long-range motion. The PMA–salt has no $f_{\max}^{Z^{\prime \prime }}$ at low salt content and one broad $f_{\max}^{Z^{\prime \prime }}$ is observed in *Z*″ spectrum of system with high salt content at frequency. This indicates no dipolar entities in PMA with low salt content and only appears after adding higher content of salt under the experimental condition.

Figure 7 displays the impedance spectra of PEO/PMA mixtures at different *W*_S_. PEO, PMA, and its blends with salt are above the glassy state of parent polymers (c.f. glass transition temperature). *R*_b_ values clearly decrease with increasing salt content for all systems and this indirectly infers the increasing DC conductivity. For system with PEO in excess, under *W*_S_ = *const*., either for low or high content of salt, the *R*_b_ decreases slightly as compared to the binary PEO/Li salt system. This may indicate that the system with PEO in excess behaves similar to PEO especially at low salt content, for instance, PEO_80_ at *W*_S_ = 0.0196 (low content of salt). A large distribution of relaxation times is observed for PEO and PEO_80_ with high salt content *W*_S_ = 0.0909 as well as for PEO_50_ at *W*_S_ = 0.0196–0.0909 and PMA at *W*_S_ = 0.0909, based on the observation of the characteristic frequencies $f_{\max}^{Z^{\prime \prime }}$ < $f_{\mathrm{cross}}^{Z^{\prime }-Z^{\prime \prime }}$ at the respective compositions. Additionally, the presence of $f_{\min}^{Z^{\prime \prime }}$, which points towards electrode polarization, may also contributes to the distribution of relaxation times of a system (i.e., PEO–PEO_50_ at *W*_S_ ≥ 0.0196 in Figure 7). This deviation may indicate to non-Debye response that normally occurs because of the increase in inhomogeneity in the system. We may summarize that, at high salt content, the distribution of characteristic frequencies is always denoted as
(9)$$f_{\max}^{Z^{\prime \prime }}<f_{\mathrm{cross}}^{Z^{\prime }-Z\prime \prime \;\;}$$
whereas at low salt content, it is seen as
(10)$$f_{\max}^{Z\prime \prime }\approx f_{\mathrm{cross}}^{Z^{\prime }-Z\prime \prime }\;\;$$

At *W*_S_ = *const*. especially at high salt content, when PMA is in excess or in symmetric systems (i.e., *W*_PEO_ ≤ 0.5) *R*_b_ and characteristic frequencies shift to higher and lower values, respectively, as compared to binary PEO/Li salt blends or system with PEO in excess. It is noted that these systems are seen as liquid–liquid phase separated as reflected in the *T*_g_ results of the same system, where situations (I) and (IV) are correlated. This behavior may indicate preferential localization of salt in the different polymer phases of the immiscible ternary mixture. For instance, based on the *T*_g_ results, we may assume that the PEO phase in PEO_50_ at *W*_S_ = 0.091 contains less salt as compared to the binary PEO/Li salt system at the same composition. This phenomenon might be one of the reasons for the enhancement of the bulk resistance as well as large deviation from Debye response in the liquid–liquid phase separated systems.

Further insight, Figure 7 illustrates that impedance spectra of PEO/PMA blends comprise low and high concentration of salt, respectively. Interesting points of the impedance spectra are suggested as follows. Blends with PEO in excess show similar behavior of neat PEO at low salt concentration. The amorphous network-like phase of PEO in the blends is almost not influenced by the PMA phase. The PEO amorphous phase contains slightly more salt than in the neat PEO; as a consequence, bulk resistance is slightly reduced. Additionally, we note that $f_{\min}^{Z^{\prime \prime }},$
$f_{\max}^{Z^{\prime \prime }}\;\left(\mathrm{PEO}\right)\;<\;f_{\min}^{Z^{\prime \prime }},$
$f_{\max}^{Z^{\prime \prime }}\;\left({\mathrm{PEO}}_{80}\right)$ and $f_{\mathrm{cross}}^{Z^{\prime }-Z^{\prime \prime }}\left(\mathrm{PEO}\right)<f_{\mathrm{cross}}^{Z^{\prime }-Z^{\prime \prime }}\left({\mathrm{PEO}}_{80}\right)$. Enhanced content of PMA in blends leads increasingly to higher *R*_b_ at *Y*_S_ = *const*. On top of that, $f_{\min}^{Z^{\prime \prime }},$
$f_{\max}^{Z^{\prime \prime }},\;\mathrm{and}$
$f_{\mathrm{cross}}^{Z^{\prime }-Z^{\prime \prime }}$ shift to lower values. It indicates “effective″ salt content of the amorphous PEO phase lessens more and more. Again, one observes more dispersion of relaxation times, $f_{\min}^{Z^{\prime \prime }}<f_{\max}^{Z^{\prime \prime }}<f_{\mathrm{cross}}^{Z^{\prime }-Z\prime \prime }$. In short, dielectric behavior of PEO and of the blends with PMA is governed by amorphous network-like phases. Addition of Li–salt leads to dissolution of low “effective″ amounts of salt in these phases. Therefore, corresponding polymer–salt mixture can be seen to good approximation as a perfect solution. This PEO-like behavior is maintained when low content PMA is added to PEO at room temperature.

Figure 8 depicts the double-logarithmic plots of conductivity (*σ*_DC_) as a function of salt concentration (*Y*_S_). The salt concentration *Y*_S_ is defined as the ratio of mass of salt to the mass of polymer. The double-logarithmic plots of *σ*_DC_ vs. *Y*_S_ for PEO/PMA/LiClO_4_ systems denote a power law dependence resulting from the functional relationship between the two quantities [27,34]. The linear relationship of the double-logarithmic plot of *σ*_DC_ vs. *Y*_S_ can be read as
(11)$$\sigma_{\mathrm{DC}}=\sigma_{o}Y_{S}^{x}$$
which yields the quantity of *σ*_o_ and allows for the estimation of the exponent *x* [27,34]. The exponent *x* in Equation (11) is the extent of association between salt molecules and the polymer segments. The quantity *σ*_o_ in Equation (11) is the simplified form of charge mobility in the system. The full form equation can be read as
(12)$$\sigma_{\mathrm{DC}}=N_{A}e\mu \alpha \frac{\rho_{P}}{M_{S}}Y_{S}^{x}$$
where *μα* is the mobility, *N*_A_ is the Avogadro’s constant, *e* is the elementary charge, *ρ*_p_ is the polymer’s density, and *M_S_* represents the molar mass of LiClO_4_, where the molecular characteristics for determination of quantity *μα* are listed in Table 1. The density of each blend was calculated using the additivity rule. All characteristic quantities of the linear regression function for all systems adopted from Equation (12) are summarized in Table 5.

From Figure 8, binary PMA/Li salt systems display the lowest *σ*_DC_ as compared to the binary PEO/Li salt blends and in the blends of PEO/PMA systems due to the fact that the *μα* mobility in PMA is by three orders of magnitude lower than in PEO. Moreover, PEO has stronger association between salt molecules and polymer segments than salt in PMA, as reflected in exponent *x* = 1.96 for PEO as compared to *x* = 1.59 for PMA, which might be the reason for lower ***σ***_DC_ of PMA. This indicates that the salts interact with PMA very weakly, which is in agreement with the *T*_g_ findings of the systems. PEO_80_ at *W*_S_ = 0.107 has the highest value of ion conductivity at 1.43 × 10^−5^ S cm^−1^ as compared to the other blend compositions with *W*_S_ = *const*. This is in good agreement where PEO_80_ also has the highest value of exponent *x* = 2.28 and *μα* = 3.81 × 10^−6^ cm^2^ V^−1^ s^−1^ as compared to the other blend compositions. The blends *W*_PEO_ = 0.5 exhibit lower values of *μα* mobility than PEO but higher than *μα* mobility of PMA. Hence, we may conclude the ionic conductivities of PEO/PMA/Li blends are dominated by the percolation of PEO in the amorphous phase. Blending small contents of PMA with PEO, when PEO in excess with addition of salt enhances the *σ*_DC_ of systems as compared to binary systems of PEO and salt with *W*_S_ = *const*. However, the conductivities of the symmetric system, i.e., PEO_50_, decrease by two orders of magnitude as compared to PEO–salt systems. This might be caused by the fact that the Li cation is not coordinating well with the chains of both polymers for the phase-separated systems at room temperature. Therefore, we may assume that when PEO with addition of salt is dispersed in PMA domains such as for systems *W*_PEO_ ≤ 0.5 where the Li–salt in the PEO phase may not have sufficient percolating paths unlike the systems with PEO in excess.

## 5. Conclusions

The influence of salt in binary PEO/PMA blends was elucidated using DSC, OM, and EIS. The changes in phase behavior indicated by *T*_g_ and morphologies are correlated to the dielectric relaxation using impedance spectroscopy. Based on *T*_g_s, PEO/PMA remains miscible when PEO is in excess after the addition of low contents of salt. However, this behavior is not seen in systems of *W*_PEO_ < 0.6 at salt content *W*_S_ > 0.05 due to the fact of preferential solubility of salt in the PEO phase, which leads to liquid–liquid phase separation. These findings are well reflected with the results from optical microscopy, where the morphologies of PEO/PMA blends with or without addition of salt can be distinguished in agreement with *T*_g_ findings. Subsequently, the evaluation of polarization and dielectric relaxation of dipoles in the composition dependent PEO/PMA blends with salts at room temperature are discussed from the phenomenological point of view. Dielectric relaxation for systems with PEO in excess behaves similar as in PEO after the addition of salt. Main dipole relaxation and bulk resistance of the salt-added systems are shifted to higher frequency and lower impedance, respectively, with increasing salt content. This phenomenon is also true for all systems under discussion. Moreover, the double-logarithmic plot of *σ*_DC_ vs. salt concentration manifests the presence of power-law distribution between the quantities. It indicates that the conductivity of the compositional-dependent PEO/PMA blends with salts are governed by the conductivity of PEO, as the power-law distribution of compositions with PEO in excess are higher than in PMA.

## Figures and Tables

**Figure 1 polymers-12-02963-f001:**
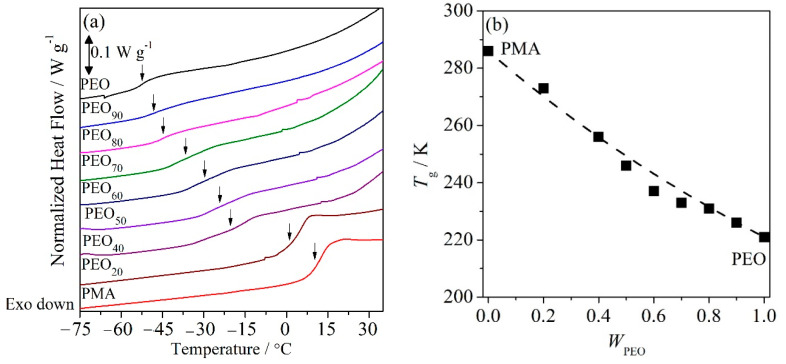
(**a**) Differential scanning calorimetry (DSC) thermograms of poly(ethylene oxide) (PEO)/poly(methyl acrylate) (PMA) blends and (**b**) variation of *T*_g_ of PEO/PMA blends versus mass fraction of PEO; the dashed line represents the *T*_g_ estimated after Fox equation [Equation (3)].

**Figure 2 polymers-12-02963-f002:**
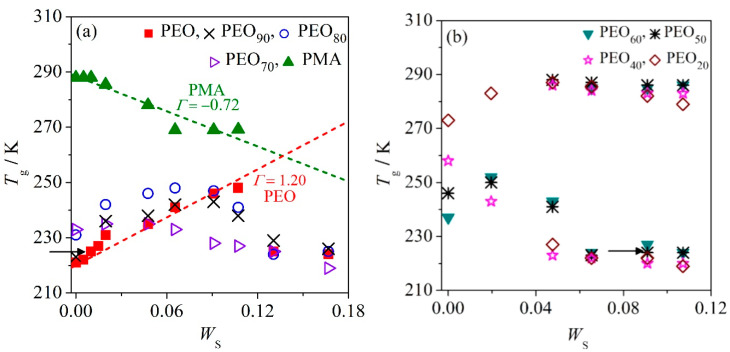
Variations of glass transition temperature (*T*_g_) of (**a**) *W*_PEO_ ≥ 0.7 with linear regression of Equation (5) and (**b**) *W*_PEO_ ≤ 0.6 versus mass fraction of salt. Refer to the text for black arrows.

**Figure 3 polymers-12-02963-f003:**
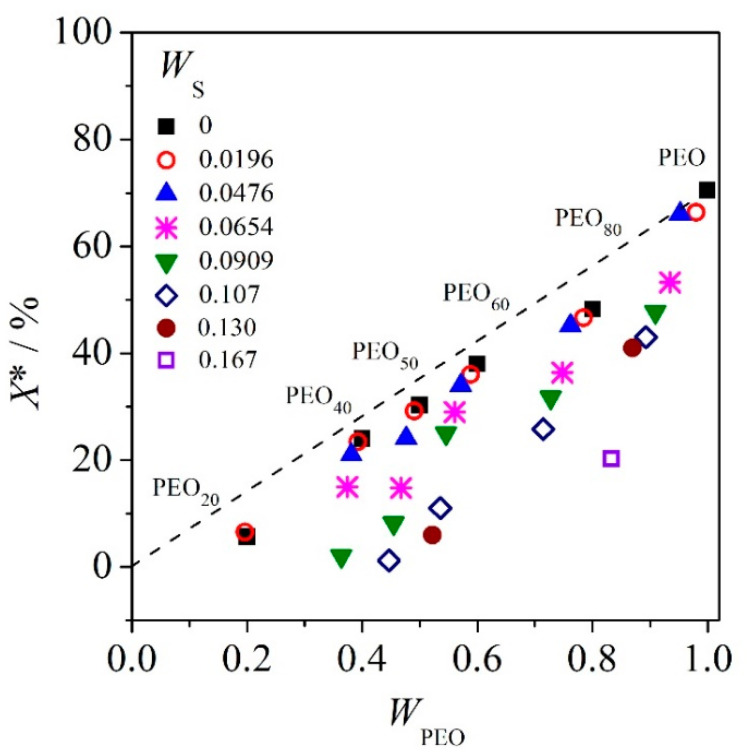
Crystallinity (*X**) of PEO in the mixture of PEO/PMA/LiClO_4_ versus mass fraction of PEO; the dashed curve indicates the loci of constancy of PEO crystallinity in PEO/PMA/LiClO_4_.

**Figure 4 polymers-12-02963-f004:**
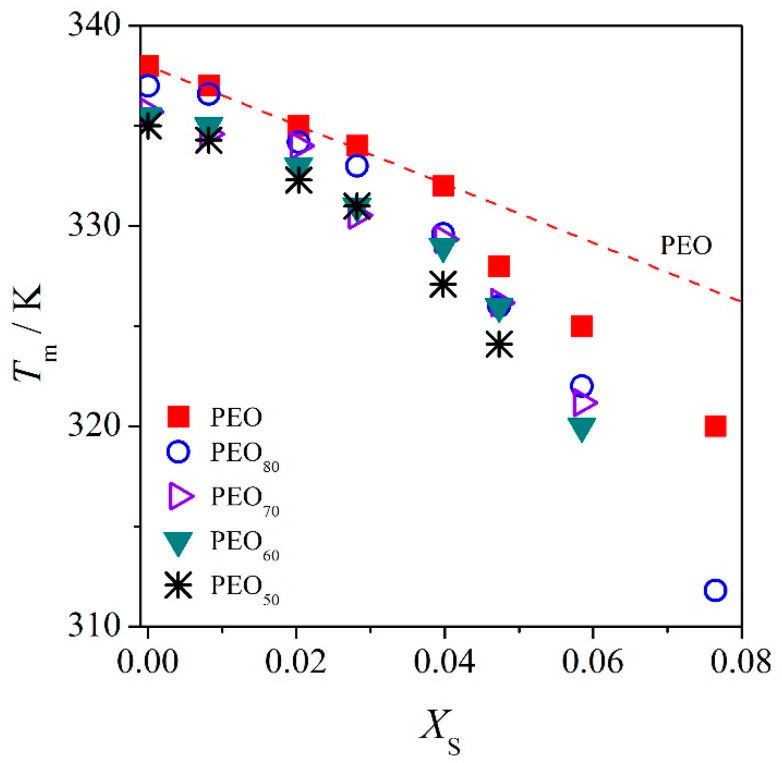
Apparent melting point (*T*_m_) of PEO in the mixture of PEO/PMA/LiClO_4_ versus mole fraction of salt; the dashed curve is the linear regression for low salt content after Equation (8). *X*_S_ represents the mole fraction of salt added to the system.

**Figure 5 polymers-12-02963-f005:**
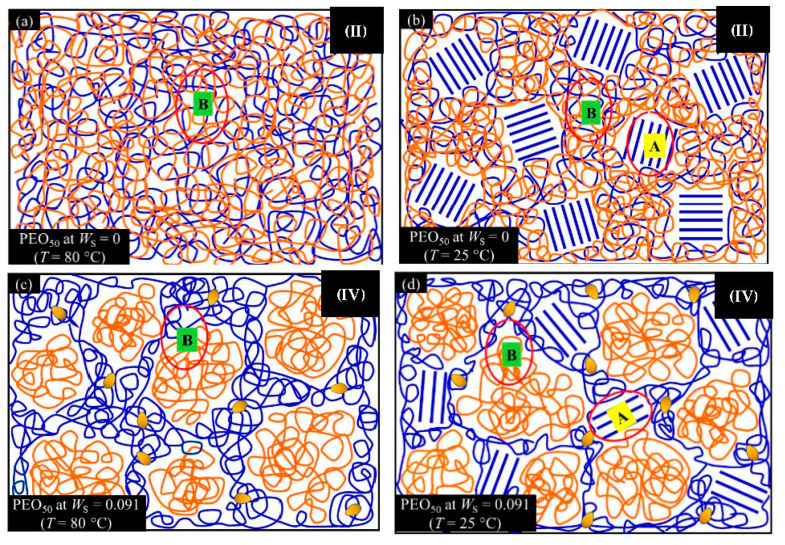
Schematic diagrams of PEO_50_ at (**a**) *W*_S_ = 0 and *T* = 80 °C; (**b**) *W*_S_ = 0 and *T* = 25 °C; (**c**) *W*_S_ = 0.091 and *T* = 80 °C; (**d**) *W*_S_ = 0.091 and *T* = 25 °C. Blue and orange polymer chains represent PEO and PMA, respectively. The crystalline phase of PEO is presented as the highly ordered chains (labelled as **A**) and the amorphous phase of PEO and PMA are presented as the entangled chains (labelled as **B**). The LiClO_4_ is presented as the yellow irregular oval-shaped entities.

**Figure 6 polymers-12-02963-f006:**
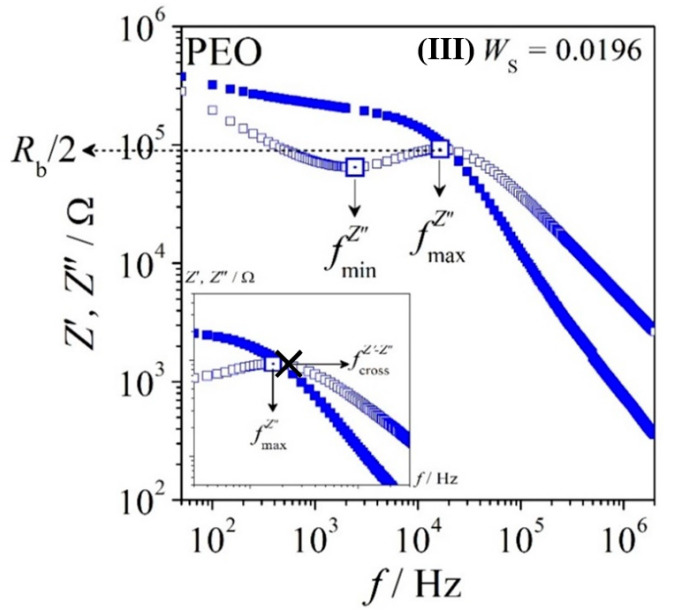
Frequency-dependent impedance spectra of PEO with mass fraction of salt 0.0196 (~2 wt.%); Solid and open markers represent the real part (*Z*’) and the imaginary part (*Z*″) of impedance, respectively.

**Figure 7 polymers-12-02963-f007:**
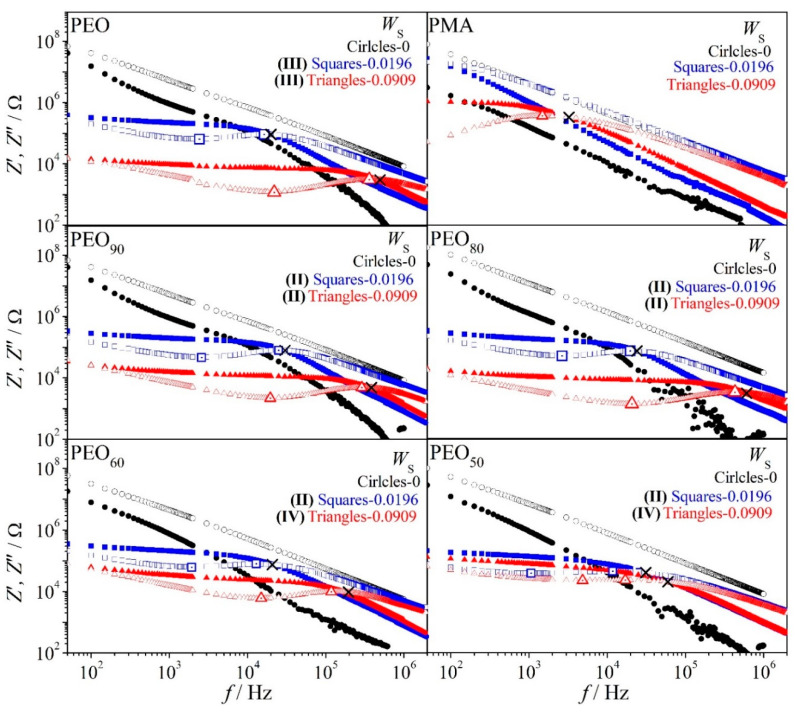
Frequency-dependent impedance spectra of PEO/PMA/LiClO_4_ systems at different *W*_S_ at 25 °C; solid and open markers represent the real part (*Z*′) and the imaginary part (*Z″*) of impedance, respectively.

**Figure 8 polymers-12-02963-f008:**
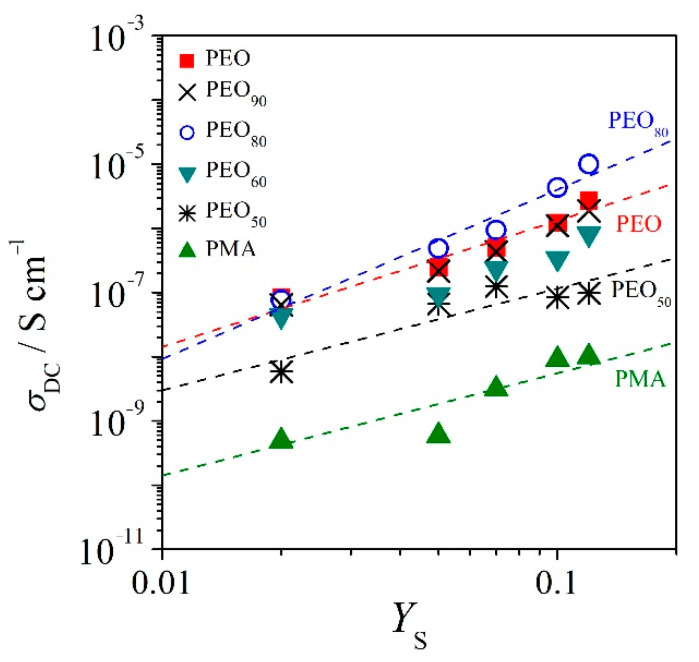
Conductivity (*σ*_DC_) of PEO/PMA/LiClO_4_ systems versus salt concentration at temperature 25 °C; the dashed curves are the linear regressions after Equation (11).

**Table 1 polymers-12-02963-t001:** Brief interpretations on the morphology of the ternary mixtures in the amorphous phase.

Situation	Possible Morphology	System
(I)	Amorphous polyethylene oxide (PEO) incorporated with Li-salt/amorphous polymethyl acrylate (PMA)	PEO/PMA blends with salt content more than ~5 wt.% when PMA is in excess
(II)	Amorphous mixture of PEO and PMA incorporated with Li–salt	PEO/PMA blends with salt content less than ~10 wt.% when PEO is in excess
(III)	Amorphous PEO incorporated with Li–salt/pure Li–salt	PEO with salt content more than ~10 wt.%
(IV)	Amorphous PEO incorporated with Li–salt/amorphous PMA/pure Li–salt	All PEO/PMA blends with salt content more than ~10 wt.%

**Table 2 polymers-12-02963-t002:** Characteristics of materials.

Characteristics	PEO	PMA	LiClO_4_
*M*_w_^(a)^/g mol^−1^	-	-	106.4
*M*_η_^(b)^/g mol^−1^	300,000	40,000	-
*T*_m_^(c)^/°C	65	-	236 ^(f)^
*T*_g_^(d)^/°C	−53	13	-
*ρ*^(e)^/g cm^−3^	1.21	1.22	2.42
Molecular Structure	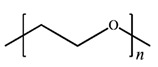	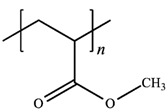	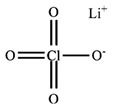
Supplier	Sigma Aldrich Chemical Co. (St. Louis, MI, USA)		Acrõs Organics Co. (Geel, Antwerp, Belgium)

^(a)^ Mass-average molar mass as determined by the supplier. ^(b)^ Viscosity-average molar mass as determined by the supplier. ^(c)^ Melting temperature of PEO estimated in this study. ^(d)^ Glass transition temperature estimated in this study. ^(e)^ Density as determined by the supplier. ^(f)^ Melting temperature from reference [22].

**Table 3 polymers-12-02963-t003:** The characteristic quantities obtained from Equations (5) and (8) for PEO/PMA/LiClO_4_ systems.

Systems	*T*_g_^o^/K	*Γ* Retrieved from Equation (5)	${\bar{{∆C}_{P}\;}}^{\left(a\right)}/$ J mol^−1^ K^−1^	Δ*G/* kJ mol^−1^	*T*_m_^o^/K	Melting Point Depression Retrieved from Equation (8) [Correlation]	Activity Coefficient (*γ*_p_)
**PEO**	220	1.20	8.9 ± 2.6	2.4	338	∆*T/*K = 147.3 *X*_S_ [0.999]	1 + 0.30 *X*_S_
**PEO_90_**	223	1.15	8.2 ± 2.5	2.1	337	∆*T/*K = 123.7 *X*_S_ [0.999]	1 + 0.37 *X*_S_
**PEO_80_**	231	1.27	9.2 ± 2.4	2.7	336	∆*T/*K = 163.7 *X*_S_ [0.999]	1 + 0.45 *X*_S_
**PEO_70_**	233	0.21	8.1 ± 1.7	0.4	335	∆*T/*K = 91.8 *X*_S_ [0.999]	1 + 0.19 *X*_S_
**PEO_60_**	237	-	-	-	335	-	-
**PEO_50_**	246	-	-	-	334	-	-
**PEO_40_**	258	-	-	-	333	-	-
**PEO_20_**	273	-	-	-	331	-	-
**PMA**	286	−0.72	43.4 ± 1.1	−8.9	-	-	-

^(a)^ Mean value of Δ*C*_P_ of the polymer pair in the linear range.

**Table 4 polymers-12-02963-t004:** Micrographs of PEO/PMA/LiClO_4_ systems at temperature of 80 °C after 30 min and 25 °C after 24 h of isothermal annealing. Micrographs were taken with 10× magnification. The scale bar at each micrograph corresponds to 100 μm.

System		80 °C	25 °C
**PEO**	*W*_S_ = 0	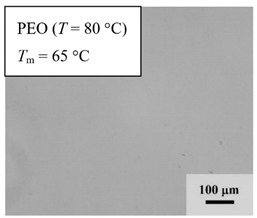	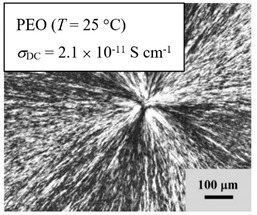
	*W*_S_ = 0.0196	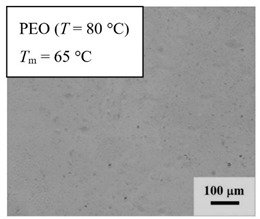	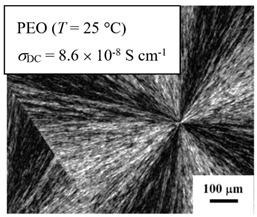
	*W*_S_ = 0.091	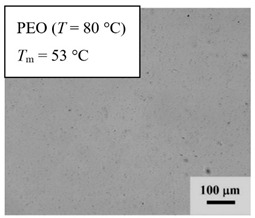	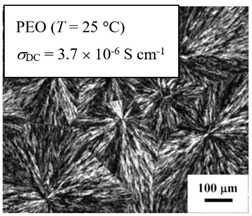
**PEO_80_**	*W*_S_ = 0	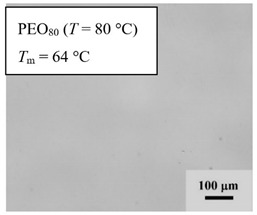	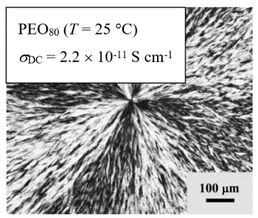
	*W*_S_ = 0.0196	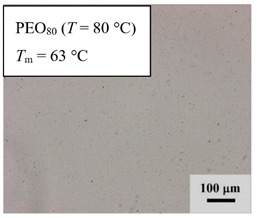	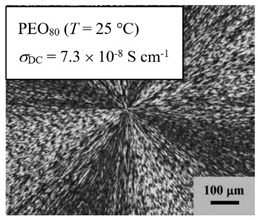
	*W*_S_ = 0.091	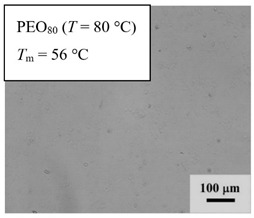	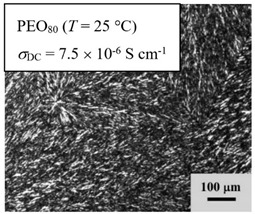
**PEO_50_**	*W*_S_ = 0	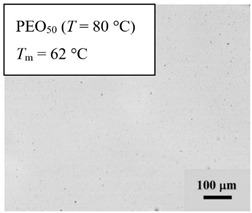	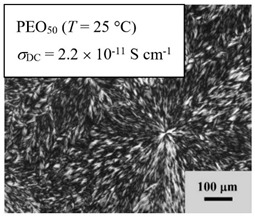
	*W*_S_ = 0.0654	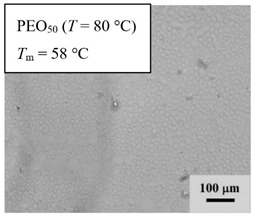	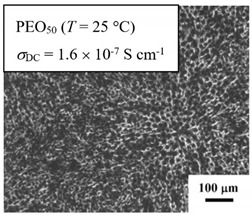
	*W*_S_ = 0.091	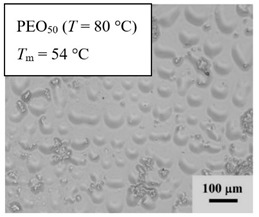	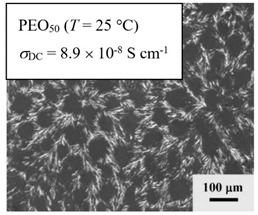

**Table 5 polymers-12-02963-t005:** The characteristic quantities obtained from Equation (12) for PEO/PMA/LiClO_4_ systems.

Systems	Regression Function/S cm^−1^	Correlation	*X*	*μα* /cm^2^ V^−1^ s^−1^
PEO	*σ*_DC_ = 2.82 × 10^−4^ *Y*_S_^1.96^	0.950	1.96	2.57 × 10^−7^
PEO_90_	*σ*_DC_ = 1.10 × 10^−4^ *Y*_S_^1.87^	0.978	1.87	1.01 × 10^−7^
PEO_80_	*σ*_DC_ = 4.19 × 10^−3^ *Y*_S_^2.28^	0.944	2.28	3.81 × 10^−6^
PEO_70_	*σ*_DC_ = 3.16 × 10^−5^ *Y*_S_^1.68^	0.901	1.68	2.87 × 10^−8^
PEO_60_	*σ*_DC_ = 2.79 × 10^−6^ *Y*_S_^1.58^	0.944	1.58	2.54 × 10^−9^
PEO_50_	*σ*_DC_ = 1.95 × 10^−6^ *Y*_S_^1.59^	0.957	1.59	1.77 × 10^−10^
PMA	*σ*_DC_ = 1.57 × 10^−6^ *Y*_S_^1.59^	0.906	1.59	1.43 × 10^−9^

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
