# Peer review of "Effects on the Properties after Addition of Lithium Salt in Poly(ethylene oxide)/Poly(methyl acrylate) Blends"

_polymers, 2020, doi:10.3390/polym12122963_

Round 1

Reviewer 1 Report

The manuscript presents an experimental work on measuring several physical properties of PEO/PMA blends and their dependence on the addition of LiClO4.

The experimental procedures are well described and the results are clearly presented, while the discussion part could be improved as currently it is more of descriptive nature than theoretical analysis. But the main issue I have with this manuscript is that it repeats an already published and cited paper by the same authors (doi:10.1002/masy.201600050). Although current manuscript contains some improvements, some new information and more data points, overall it repeats the published paper by more than 50% . I suggest the authors to reorganize the structure of the present manuscript, clearly state that this is a continuation and improvement of the previous work and focus the discussion section on the improvements and new findings in comparison to the published work.

Small remark in this regard – it would be beneficial to keep the plots in 2 papers with same units and same symbols. For example, temperature was plotted in Celsius and now is in Kelvin. Mass fractions were denoted with Y and is W, etc.

Several other comments:

1) The Introduction section is well written and sufficiently supports the motivation for the work. It would be helpful for the reader if the 4 different “situations” for the morphology of the ternary blends were summarized in some schematic representation. The authors refer to these “situations” throughout the manuscript, so it would be beneficial to have a one-glance reference instead of the full page of text.

2) The excessive use of parentheses for comments is tiresome and in most cases unnecessary. The remarks can either be included in the main text or completely omitted. There are a lot of repetition throughout the text. For example, some parts of the experimental section are repeated in the results section without the need to do so. These issues make the manuscript unreasonably long to read through.

3) As I mentioned above, the discussion section is mostly an observation of the obtained results with little to no in-depth attempts to physical explanation of the observed phenomena. It is still valuable information, but insufficient for a scientific research.

Minor remarks:

The section numbering is broken, eq. 8 format is different and the formula on line 332 is repeated twice.

Author Response

Thank you for the comments. The responses are listed in the attached PDF file. Please see the attachment.

Reviewer 2 Report

Halim et al. designed PMA/PEO mixtures with different mass ratios, and explored the thermodynamic and electrochemical performances after adding different contents of lithium salt, at the same time, they used many thermodynamic calculations and analyses which is logically rigorous and interesting. However, some phenomena are unclear without further justification. Therefore, I recommend publication after addressing the following the questions. 

  1. Pg 6Question 1:figure2a,the Tg of PMA decreases with elevating salt content (WS)”the. The author explained that it is because of the solubility of lithium salt, which is not rigorous. We observed that it has a gentle upward trend at 0.06-0.12Ws, which indicates that low content of lithium salt may inhibit the formation of amorphous. Moreover this is a form of Li salt intercalation. Why didn't the author use in-situ XRD to analyze the dissolved amount or intercalation amount of Li salt, and only use the design quality to represent the Li salt content? Question 2: figure2b, Why do PEO20, PEO40 and PEO60 exit two curves at 0.04-0.12Ws? 
  1. Pg 11Table 3. If you want to distinguish the crystalline phase from the amorphous phase more clearly, and want to prove that the Li salt is concentrated in the amorphous phase, why don't the authors use high-resolution SEM to observe more details, and use EDS to analyze the distribution of Li salt. 
  1. Pg 16figure 7. What is the open circuit potential that the author inputs when testing EIS? The impedance maps corresponding to different initial potentials are completely inconsistent.
  1. Pg 17figure 8. As we all know, the more ion doping, the higher the conductivity, so what is the highlight of this article? The author should tell us from the existence form and mechanism of Li salt in the PMA/PEO system how this is different from "ion doping", and it requires more experimental data to support.

Author Response

(The authors gave the same response as above.)

Reviewer 3 Report

The proposed manuscript demonstrates a number of experimental measurements.
Unfortunately, I found no interpretations and explanations of obtained observations. The sentence in the end of Conclusions "the morphologies of the systems and molecular interaction of polymer chains with salt molecules may rule the dielectric or electric relaxation of the systems", being absolutely correct, seems trivial.

Nevertheless, all measurement, presented in the manuscript, are done accurate and correct; the manuscript contains no incorrect results. Therefore, the manuscript can be recommended for publication after minor revision by adding some explanations of obtained experimental results.

Author Response

(The authors gave the same response as above.)

Round 2

Reviewer 1 Report

The manuscript was considerably improved, the authors addressed most of my comments. I am still not convinced that the manuscript differs enough from its predecessor to be published as a separate paper, but I leave this decision to the editor.
Minor style remarks:
1) The comma after the equation and before the word "where" should exist but it should be on the same line as formula, it is not correct to start a line with a comma, or to omit it completely. (see lines 135, 175, 248, 322, 495, 500)
2) Excessive use of parentheses is still there. I guess, it could be regarded as authors style (but I still think that it is an unnecessary struggle for the reader, especially when it is the whole sentence in parentheses). ( see lines 38, 44, 49, 53, 57, 59, 79, 84, 85, 105, 108, 117, 132, 130, 133, 156, 162, 169, 186, 210, 245, 266, 271, 272, 273, 296, 369 and so on)

Reviewer 2 Report

The revised manuscript is now suitable for publication.